# SgCG: Semantic-Guided Contrastive Generalization for Medical Image Segmentation

## Abstract

After training on the source domain, deep learning models often struggle to generalize effectively to unknown target domains with differing data distributions. This is an even more severe challenge when the target domain is not available. In this paper, we tackle the problem of domain-generalized medical image segmentation by introducing a novel semantic-guided contrastive generalization algorithm, termed SgCG. The method aligns different multi-source domains based on semantic distributions to learn domain-invariant features. Specifically, we implement a novel contrastive generalization loss at the pixel level that incorporates semantic distributions from the source domains. This approach facilitates the clustering of pixel representations from the same category while effectively separating those from different categories, thereby improving the model's segmentation performance while learning domain-invariant features. Furthermore, we establish an upper bound estimation for the SgCG approach by integrating a contrastive generalization loss which includes an infinite number of both similar and dissimilar pixel pairs. Despite the simplicity and straightforwardness of the approach, our empirical analysis reveals mechanisms that can maximize the potential of SgCG. We demonstrate the effectiveness of our approach using two public benchmarks for generalizable segmentation in medical images, where it achieves state-of-the-art performance.

## 1 Introduction

Image segmentation Wu et al. (2024), a long-standing research focus in computer vision, poses a core challenge in medical image analysis. Within this domain, tasks may involve various imaging techniques, including microscopic examination Schoch & Maywald (1956), Computed Tomography (CT) Buzug (2011), X-rays Hessenbruch (2002) and Magnetic Resonance Imaging (MRI) Katti et al. (2011). They can span different biomedical areas, such as retinal imaging, brain, thoracic, abdominal, or even individual cells, and may target diverse structures like cardiac valves or ventricles. This variety has led to the development of numerous specialized segmentation tools, each optimized for specific tasks or closely related sets of tasks. In recent years, deep learning models have emerged as the dominant approach for medical image segmentation Ouyang et al. (2022); Azad et al. (2024); Song et al. (2022); Zhang et al. (2023), driving significant advancements in the field.

In the field of image segmentation Minaee et al. (2021), a major challenge is how to overcome the performance drop of models when faced with out-of-distribution samples. This issue is particularly prominent in the medical domain, as clinical researchers and other scientists continuously define new segmentation tasks based on evolving population characteristics, scientific advancements, and clinical objectives. However, domain adaptation Ben-David et al. (2006); You et al. (2019) requires access to target domain data, which is often difficult to obtain in real-world scenarios due to privacy concerns associated with medical data. Compared to domain adaptation, domain generalization Wang et al. (2021b) is a more general task that only requires training a labeled segmentation model on the source domain, allowing it to generalize to unseen target domain data. The difficulty of domain generalization in image segmentation tasks lies in the fact that, during training, target domain data is typically unavailable, making it challenging for the model to learn the characteristics of the target domain. A common solution is to enhance the diversity of training data through image-level Luo et al. (2021); Chen et al. (2017) data augmentation. In simple terms, by expanding the source domain dataset to include more representative diverse samples, the model's ability to

generalize to unknown domains can be improved, especially when only limited source domain data is available. However, current methods for increasing diversity mainly focus on transformations in the image space, making it more complex to generate images in new domains, as it is difficult to specify or learn effective image synthesis strategies.

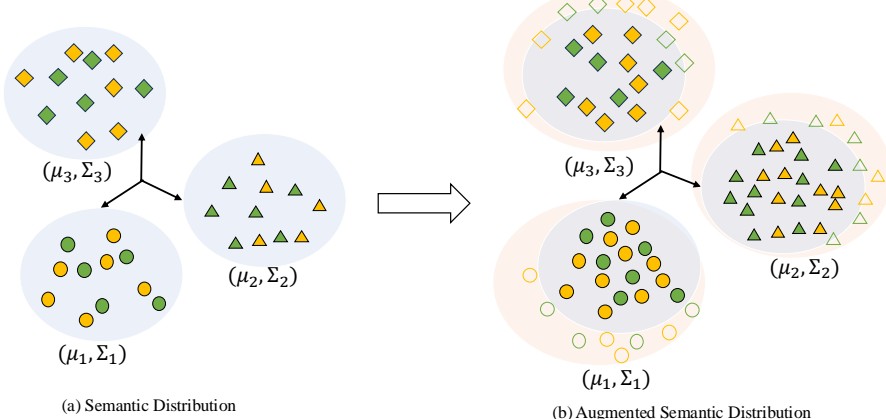

(a) Semantic Distribution  (b) Augmented Semantic Distribution

Figure 1: We sample directions from a zero-mean multivariate normal distribution with the estimated covariance as the variance, and apply them to the features of the training samples in that class.

Inspired by implicit semantic data augmentation algorithms Wang et al. (2021a), we propose a new perspective for addressing the domain generalization problem in semantic segmentation. Our method learns domain-invariant features by attracting similar pixels and repelling dissimilar pixels in a pixel-wise representation, thereby reducing domain shift. We first estimate the covariance matrix online for each category, capturing intra-class variation. Then, we sample directions from a zero-mean multivariate normal distribution with the estimated covariance as variance, applying these directions to the features of training samples from that category (see Figure. 1). This approach generates diverse samples from the estimated distributions, enhancing source domain data. Secondly, we observe that enhancing the intra-class compactness and inter-class separability of pixel representations can significantly improve the performance of dense pixel classifiers. Therefore, we separate pixel-wise representations in both the source and target domains, implicitly defining an infinite number of positive sample pairs for each pixel by sampling from the estimated distributions of the same category. Based on this, we design a pixel-level special contrastive loss function for contrastive adaptation. We also derive an upper bound for this loss function, ensuring its effectiveness in practical applications. In essence, this represents a novel and robust alternative loss function. Since explicit data samples do not need to be generated, we term our algorithm semantic-guided contrastive generalization (SgCG). We recommend using the popular framework RAM Zhou et al. (2022b) as a baseline to validate the effectiveness of our method SgCG. Experimental results demonstrate that aligning source and target pixel representations with semantic distributions through contrastive learning can effectively reduce domain discrepancies and enhance generalization capability in the target domain. In summary, our main contributions can be summarized as follows:

- We propose a novel semantic-guided contrastive generalization (SgCG) for medical image segmentation, which encourages the source feature augmentations in an implicit manner.

- By deriving the upper bound of the expected contrastive loss using statistical data from the distribution of each category, we enable the learning of invariant and distinctive pixel-wise representations to be both straightforward and effective.

- Extensive empirical evaluations on several competitive benchmarks, including Fundus Wang et al. (2020) and Prostate Liu et al. (2020), demonstrate that SgCG significantly improves the baseline model. Additionally, analytical evidence is provided to validate its effectiveness.

## 2 METHOD

### 2.1 DEFINITION AND OVERVIEW

We define a set of $K$ source domains, denoted as $D_s$, where each source domain contains $N_k$ pairs of images and their corresponding segmentation labels, represented as $D_s = \left\{ \left( x_i^k, y_i^k \right)_{i=1}^{N_k} \right\}_{k=1}^{K}$. Here, $x_i^k$ refers to the $i$-th image in the $k$-th source domain, while $y_i^k$ is the segmentation label for that image. Our goal is to develop a medical image segmentation model $F_\theta$ that can be trained using the source domain dataset $D_s$ and possesses good generalization ability. We aim for the model $F_\theta$ to perform well on unseen target domain $D_t$, where $D_t = \{x_i\}_{i=1}^{N_t}$, with $x_i$ being the $i$-th image in the target domain and $N_t$ representing the number of images in the target domain.

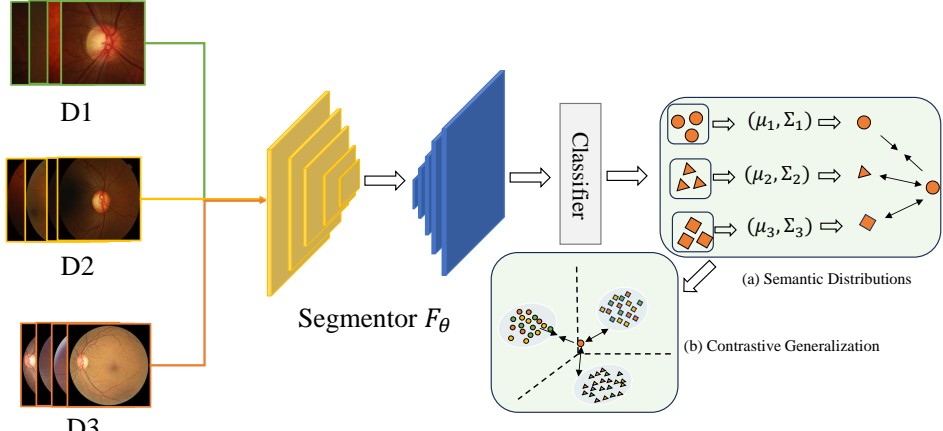

Figure 2: Framework of SgCG. By using semantic-guided contrastive generalization for comparative matching of different semantics, features with the same semantic concept will be brought closer together, while features with different semantic concepts will be pushed apart across domains.

Our proposed Semantic-guided Contrastive Generalization (SgCG) method for medical image segmentation is illustrated in Figure 2. In front of our training pipeline, we introduce a self-supervision domain-specific image restoration encoder-decoder module as our baseline Zhou et al. (2022b). Then, the encoder-decoder segmentation model is trained by the semantic-guided contrastive generalization loss of the source domain images, which can cluster of pixel representations from the same category are obliged to cluster together and those from different categories are obliged to spread out, boosting segmentation capability of the model. Finally, an upper bound on this formulation is derived by implicitly involving the simultaneous learning of an infinite number of (dis)similar pixel pairs, making it highly efficient.

### 2.2 IMPLICIT SEMANTIC DATA AUGMENTATION

Most traditional data augmentation in image segmentation methods Zhao et al. (2019); Xie et al. (2023); Chen et al. (2022) make modifications directly on training images. In contrast, ISDA performs data augmentation at the feature level which translating image features along meaningful semantic directions. Such directions are determined based on the covariance matrices of deep features. Specifically, for $C$-class classification problem, ISDA statistically estimates the class-wise covariance matrices $\hat{\Sigma} = \left\{ \hat{\Sigma}_1, \hat{\Sigma}_2, \dots, \hat{\Sigma}_C \right\}$ in an online manner at each training iteration. For the arbitrary source pixel $i \in \{1, 2, \cdots, H' \times W'\}$ in $\hat{F}_s$, we sample transformation directions from the Gaussian distribution $\mathcal{N}\left(0, \lambda\hat{\Sigma}_{y_i}\right)$ to get the augmented features. The mean of features from the $k^{th}$ category is calculated as the average values of every single dimension in the feature vector,

$$\mu'^k = \frac{1}{|\Lambda^k|} \sum_{i \in \{1, 2, \cdots, H' \times W'\}} \mathbb{1}_{[M_{s,i}=k]} \hat{F}_{s,i}, \tag{1}$$

where $|\cdot|$ is the cardinality of the set, $\Lambda^k$ according to its mask $M_{s,i} \in \mathbb{R}^{H' \times W'}$ downsampled from ground truth label. $\hat{F}_{s,i} \in \mathbb{R}^A$ is feature representation of source pixel $i$.

For semantic-guided contrastive generalization information, we require either global category prototypes or local category centroids. On the one side, we opt for an online fashion on the entire source domain, aggregating mean statistics one by one to build global category prototypes. Mathmatically, the online estimate algorithm for the mean of the $k^{th}$ category is given by:

$$\mu_{(t)}^k = \frac{n_{(t-1)}^k \mu_{(t-1)}^k + m_{(t)}^k {\mu'}_{(t)}^k}{n_{(t-1)}^k + m_{(t)}^k}, \tag{2}$$

As the feature vector $\hat{F}_{s,i}$ is multi-dimensional, we use covariance for a better representation of the variance between any pair of elements in the feature vector. The covariance matrix $\Sigma^k$ for class $k$ is calculated as:

$$\Sigma_{(t)}^k = \frac{n_{(t-1)}^k \Sigma_{(t-1)}^k + m_{(t)}^k {\Sigma'}_{(t)}^k}{n_{(t-1)}^k + m_{(t)}^k}$$
$$+ \frac{n_{(t-1)}^k m_{(t)}^k \left(\mu_{(t-1)}^k - {\mu'}_{(t)}^k\right)\left(\mu_{(t-1)}^k - {\mu'}_{(t)}^k\right)^\top}{\left(n_{(t-1)}^k + m_{(t)}^k\right)^2}, \tag{3}$$

where ${\Sigma'}_{(t)}^k$ is the covariance matrix of the features between the $k^{th}$ category in the $t^{th}$ image. It is noteworthy that $K$ mean vectors and $K$ covariance matrices are initialized to zeros. During training, we dynamically update these statistics using Eq. 2 and Eq. 3 with source feature map from the encoder-decoder network. The estimated semantic-guided contrastive generalization of semantic statistics are more informative to guided the pixel representation learning of the source domains.

## 2.3 SEMANTIC-GUIDED CONTRASTIVE GENERALIZATION

In medical image segmentation, a few methods Wu et al. (2022); Chaitanya et al. (2020) utilize the centroids of classification features as anchors to reduce domain shift, yielding promising results. However, no one has attempted to consider the distances between features of different categories within the source domain. Building on this, we designed a semantic-guided contrastive generalization approach aimed at learning similar/dissimilar pairs at the pixel level. This method seeks to learn domain-invariant features to mitigate domain gaps through centroid-aware pixel contrast or distribution-aware pixel contrast. We take an infinity limit on the number of $M$ and $N$, where the effect of $M$ and $N$ is hopefully absorbed in a probabilistic way. With this application of infinity limit, the statistics of the data are sufficient to achieve the same goal of multiple pairing. Mathematically, as $M$ and $N$ goes to infinity, $\mathcal{L}^{M,N}$ becomes the estimation of:

$$\mathcal{L}_i^\infty = \lim_{\substack{M \to \infty \\ N \to \infty}} \mathcal{L}_i^{M,N}$$
$$= -\mathbb{E}_{\substack{q^+ \sim p(q^+) \\ q_j^- \sim p(q_j^-)}} \log \frac{e^{q_i^\top q^+ / \tau}}{e^{q_i^\top q^+ / \tau} + \sum_{j=1}^{K-1} e^{q_i^\top q_j^- / \tau}}, \tag{4}$$

where $p(q^+)$ is the positive semantic distribution that has the same semantic label and $p(q_j^-)$ is the $j^{th}$ negative semantic distribution that has different semantic label with respect to $q_i$. The analytic form of Eq. 4 itself is intractable, but Eq. 4 has a rigorous closed form of upper bound, which can

be derived as:

$$-\mathbb{E}_{q^+,q_j^-}\log\frac{e^{q_i^\top q^+/\tau}}{e^{q_i^\top q^+/\tau}+\sum_{j=1}^{K-1}e^{q_i^\top q_j^-/\tau}}$$

$$=\mathbb{E}_{q^+}\left[\log\left[e^{\frac{q_i^\top q^+}{\tau}}+\sum_{j=1}^{K-1}\mathbb{E}_{q_j^-}e^{\frac{q_i^\top q_j^-}{\tau}}\right]\right]-\mathbb{E}_{q^+}\left[\frac{q_i^\top q^+}{\tau}\right] \tag{5}$$

$$\leq\log\left[\mathbb{E}_{q^+}\left[e^{\frac{q_i^\top q^+}{\tau}}+\sum_{j=1}^{K-1}\mathbb{E}_{q_j^-}e^{\frac{q_i^\top q_j^-}{\tau}}\right]\right]-q_i^\top\mathbb{E}_{q^+}\left[\frac{q^+}{\tau}\right] \tag{6}$$

$$=\log\left[\mathbb{E}_{q^+}e^{\frac{q_i^\top q^+}{\tau}}+\sum_{j=1}^{K-1}\mathbb{E}_{q_j^-}e^{\frac{q_i^\top q_j^-}{\tau}}\right]-q_i^\top\mathbb{E}_{q^+}\left[\frac{q^+}{\tau}\right] \tag{7}$$

$$=\bar{\mathcal{L}}_i \tag{8}$$

where the inequality Eq. 6 follows form the Jensen's inequality on concave functions, i.e., $\mathbb{E}\log(X)\leq\log\mathbb{E}[X]$. To facilitate our formulation, we need some further assumptions on the feature distribution. Specifically, we assume that $q^+\sim\mathcal{N}(\mu^+,\Sigma^+)$ and $q_j^-\sim\mathcal{N}(\mu_j^-,\Sigma_j^-)$, where $\mu^+$ and $\Sigma^+$ are respectively the statistics i.e., mean and covariance matrix, of the positive semantic distribution for $q$, $\mu_j^-$ and $\Sigma_j^-$ are respectively the statistics of the $j^{th}$ negative distribution.

For any random variable $x$ that follows Gaussian distribution $x\sim\mathcal{N}(\mu,\Sigma)$, where $\mu$ is the expectation of $x$, $\Sigma$ is the covariance matrices of $x$, we have the moment generation function Wang et al. (2021a) that satisfies:

$$\mathbb{E}\left[e^{a^\top x}\right]=e^{a^\top\mu+\frac{1}{2}a^\top\Sigma a}. \tag{9}$$

Under the Gaussian assumption $q^+\sim\mathcal{N}(\mu^+,\Sigma^+),q_j^-\sim\mathcal{N}(\mu_j^-,\Sigma_j^-)$, along with Eq. 9, we find that Eq. 7 for a certain pixel representation $q_i$ immediately reduces to:

$$\bar{\mathcal{L}}_i=\log\left[e^{\frac{q_i^\top\mu^+}{\tau}+\frac{q_i^\top\Sigma^+q_i}{2\tau^2}}+\sum_{j=1}^{K-1}e^{\frac{q_i^\top\mu_j^-}{\tau}+\frac{q_i^\top\Sigma_j^-q_i}{2\tau^2}}\right]-\frac{q_i^\top\mu^+}{\tau} \tag{10}$$

$$=-\log\frac{e^{\frac{q_i^\top\mu^+}{\tau}+\frac{q_i^\top\Sigma^+q_i}{2\tau^2}}}{e^{\frac{q_i^\top\mu^+}{\tau}+\frac{q_i^\top\Sigma^+q_i}{2\tau^2}}+\sum_{j=1}^{K-1}e^{\frac{q_i^\top\mu_j^-}{\tau}+\frac{q_i^\top\Sigma_j^-q_i}{2\tau^2}}}+\frac{q_i^\top\Sigma^+q_i}{2\tau^2}. \tag{11}$$

The overall loss function with regard to each feature map in the source and target domain thereby boils down to the closed form whose gradients can be analytically solved for:

$$\mathcal{L}_{feat}=\frac{1}{|F_s|}\sum_{i\in F_s}\bar{\mathcal{L}}_i+\frac{1}{|F_t|}\sum_{j\in F_t}\bar{\mathcal{L}}_j, \tag{12}$$

where $|F_s|$ and $|F_t|$ are respectively numbers of pixels in $F_s$ and $F_t$. Based on this, an effective semantic distribution-aware contrastive loss is yielded to mitigate domain discrepancy via learning discriminative pixel representations.

## 2.4 TRAINING PROCEDURE

Aside from applying contrastive adaptation to the penultimate feature maps of the network, we training a popular method RAM Zhou et al. (2022b) as a baseline which design a domain-specific image restoration module and random amplitude mixup module for medical image segmentation to further validate the effectiveness of the proposed method. Overall, we can formulate our whole framework as a multi-task learning paradigm. The total training loss are as follows:

$$\mathcal{L}=\mathcal{L}_{ram}+\lambda\mathcal{L}_{feat} \tag{13}$$

where $\lambda$ controls the trade-off between the RAM loss and the contrastive generalization loss. The pseudo-label is illustrated in Algorithm. 1.

---

**Algorithm 1: SgCG algorithm.**

---

**Input:**

1) The pre-trained U-Net network.

2) The source domain $D_s = \left\{ \left( x_i^k, y_i^k \right)_{i=1}^{N_k} \right\}_{k=1}^{K}$.

3) The maximum iterations $K$, and hyper-parameters $\lambda$.

1 Initialize statistics $\{\mu^k\}_{k=1}^{K}$ and $\{\Sigma^k\}_{k=1}^{K}$ using $\mathcal{S}$.

3 **for** $l \leftarrow 0$ to $L$ **do**

5      **for** $k \leftarrow 0$ to $K$ **do**

7          Randomly sample a batch source image $I_s$ with $Y_s$ from $\mathcal{S}$.

9          Compute the feature maps $F_s$, segmentation outputs $O_s$, $O_t$ and pixel-level prediction $P_s$.

11          Estimate current mean values $\{\mu_{(t)}^k\}_{k=1}^{K}$ via Eq. 2 and covariance matrices $\{\Sigma_{(t)}^k\}_{k=1}^{K}$ via Eq. 3.

12      **end**

14      Separate pixel-wise representations of all source domains $D_s$ in the feature space and output space according to their masks $M_s$.

16      Train $\Phi_\theta$ using losses $\mathcal{L}_{ram}$ and $\mathcal{L}_{feat}$.

17 **end**

18 **return** $\Phi_\theta$

---

## 3 EXPERIMENTS

### 3.1 EXPERIMENT DATASETS

We evaluated our method on two public domain generalization medical image segmentation datasets including **Fundus** Wang et al. (2020) and **Prostate** Liu et al. (2020).

**Fundus** dataset contains retinal fundus images from four different medical centers, primarily used for optic cup and disc segmentation. Each domain has been divided into training and test sets. During the preprocessing stage, we followed previous studies Zhou et al. (2022b) to center-crop all images in the Fundus dataset, using a bounding box of $800 \times 800$. Subsequently, we randomly resized and cropped each image to obtain a $256 \times 256$ region as input for the network. We will train our model on the training set of the source domain and evaluate it on the testing set of the target domain.

**Prostate dataset** collected T2-weighted MRI prostate images from six different data sources, specifically for prostate segmentation. All images were cropped to the 3D prostate region, and the 2D slices in the axial plane were resized to $384 \times 384$. During model training, we fed the 2D slices of prostate images into our model and normalized the data for both datasets individually to intensity values in the range of $[-1, 1]$.

### 3.2 EVALUATION

For evaluation, we employ commonly used metrics: the Dice coefficient (Dice) and average surface distance (ASD) to quantitatively assess the segmentation results for both the overall area and surface shape. A higher Dice coefficient indicates better performance, while ASD has the opposite implication. To mitigate randomness, we conducted three repeated experiments and report the average performance.

### 3.3 IMPLEMENTATION DETAILS

We chose RAM Zhou et al. (2022b) as our baseline model and have adopted a U-Net-based encoder-decoder architecture for our segmentation framework. Our segmentation decoder is designed to learn domain-invaraint feature, closely mirroring the RAM decoder. The model train for 400 epochs on the Fundus dataset and 200 epochs on the prostate dataset, utilizing a batch size of 8 for each dataset. To optimize our model, we employed the Adam optimizer, initiating the learning rate at 0.001.

Furthermore, to ensure a stable training process, we introduced a polynomial decay schedule for the learning rate adjustments. Consistent with the original paper, we retained the same experimental settings for all other models and baselines.

## 3.4 COMPARATIVE EXPERIMENTS

**Comparison on Fundus**: We start by conducting comparative experiments on the Funds dataset. The results of the Dice coefficient for various models are shown in Table 1, while the experimental findings for the Average Surface Distance (ASD) are presented in Table 2. By observing the results, it is evident that SgCG (Ours) outperforms all baseline models (SOTA) in terms of generalization performance across each domain. In the Dice coefficient metrics presented in Table 1, SgCG (Ours) averages 0.61% higher than RAM (baseline), with improvements of 1.17%/0.17% in Domain 1, and 1.58%/0.96% in Domain 2. In the ASD metrics shown in Table 2, smaller values indicate better performance; our model similarly achieves higher generalization across nearly every domain compared to all models, resulting in an average ASD reduction of 0.78% compared to SOTA. These findings demonstrate that our model exhibits strong generalization capabilities on the Funds dataset, outperforming all current models.

Table 1: Dice coefficient of different methods on Fundus segmentation task (%). We mark the top results in bold.

| Task | Optic Cup/Disc Segmentation | | | | |
|---|---|---|---|---|---|
| Unseen Site | Domain 1 | Domain 2 | Domain 3 | Domain 4 | Avg. |
| Source | 81.44/95.52 | 77.20/87.96 | 85.11/94.56 | 72.30/90.97 | 85.63 |
| JiGen Carlucci et al. (2019) | 82.45/95.03 | 77.05/87.25 | 87.01/94.94 | 80.88/91.34 | 86.99 |
| BigAug Zhang et al. (2020) | 77.68/93.32 | 75.56/87.54 | 83.33/92.68 | 81.63/92.20 | 85.49 |
| SAML Liu et al. (2020) | 83.72/95.03 | 77.68/87.57 | 84.20/94.49 | 82.08/92.78 | 87.19 |
| FedDG Liu et al. (2021a) | 81.72/95.62 | 77.87/88.71 | 83.96/94.83 | 81.90/93.37 | 87.25 |
| DoFE Wang et al. (2020) | 84.17/94.96 | 81.03/89.29 | 86.54/91.67 | 87.28/93.04 | 88.50 |
| RAM Zhou et al. (2022b) | 85.48/95.75 | 78.82/89.43 | 87.44/94.67 | 85.84/94.10 | 88.94 |
| SgCG | **86.65/95.92** | **80.40/90.39** | **87.48/95.07** | **86.27/94.18** | **89.55** |

Table 2: Average Surface Distance (ASD) of different methods on Fundus segmentation task (voxel). We mark the top results in bold.

| Task | Optic Cup/Disc Segmentation | | | | |
|---|---|---|---|---|---|
| Unseen Site | Domain 1 | Domain 2 | Domain 3 | Domain 4 | Avg. |
| JiGen Carlucci et al. (2019) | 18.57/9.43 | 17.29/19.53 | 9.15/6.99 | 15.84/12.14 | 13.62 |
| BigAug Zhang et al. (2020) | 22.61/12.53 | 17.95/17.64 | 11.48/10.33 | 11.57/9.36 | 14.18 |
| SAML Liu et al. (2020) | 17.08/9.01 | 16.72/18.63 | 10.87/7.87 | 16.28/8.64 | 13.14 |
| FedDG Liu et al. (2021a) | 18.57/7.69 | 15.87/16.93 | 11.09/7.28 | 10.23/7.51 | 11.90 |
| DoFE Wang et al. (2020) | 16.07/7.78 | 13.44/17.06 | 10.12/10.75 | 8.14/7.29 | 11.26 |
| RAM Zhou et al. (2022b) | 16.05/7.12 | 14.01/13.86 | 9.02/7.11 | 8.29/7.06 | 10.32 |
| SgCG | **14.62/7.31** | **12.96/12.21** | **8.46/6.57** | **7.54/6.64** | **9.54** |

Table 3: Dice coefficient of different methods on Prostate segmentation task (%). We mark the top results in bold.

| Task | Prostate Segmentation | | | | | | |
|---|---|---|---|---|---|---|---|
| Unseen Site | Domain 1 | Domain 2 | Domain 3 | Domain 4 | Domain 5 | Domain 6 | Avg. |
| Source | 85.30 | 87.56 | 82.33 | 87.37 | 80.49 | 81.40 | 84.04 |
| JiGen Carlucci et al. (2019) | 85.45 | 89.26 | 85.92 | 87.45 | 86.18 | 83.08 | 86.22 |
| BigAug Zhang et al. (2020) | 85.73 | 89.12 | 84.49 | 88.02 | 81.95 | 87.63 | 86.19 |
| SAML Liu et al. (2020) | 85.88 | 88.72 | 85.03 | 88.44 | 86.72 | 87.56 | 87.05 |
| FedDG Liu et al. (2021a) | 86.12 | 89.24 | 85.30 | 88.95 | 85.93 | 86.65 | 87.03 |
| DoFE Wang et al. (2020) | **88.89** | 87.88 | 85.08 | 89.06 | 86.15 | 87.03 | 87.34 |
| RAM Zhou et al. (2022b) | 87.56 | 89.35 | 86.88 | 87.34 | **86.98** | 88.02 | 87.68 |
| SgCG | 87.96 | **90.42** | **87.23** | **89.17** | 86.78 | **88.32** | **88.34** |

**Comparison on Prostate**: To further demonstrate the generalization and robustness of SgCG (Ours) in medical image segmentation, we conduct comparative experiments on the Prostate dataset. It is

Table 4: Average Surface Distance (ASD) of different methods on Prostate segmentation task (voxel). We mark the top results in bold.

| Task | Prostate Segmentation | | | | | | |
|------|----------|----------|----------|----------|----------|----------|------|
| Unseen Site | Domain 1 | Domain 2 | Domain 3 | Domain 4 | Domain 5 | Domain 6 | Avg. |
| Source | 1.22 | 1.95 | 4.68 | 1.51 | 3.95 | 4.23 | 2.92 |
| JiGen Carlucci et al. (2019) | 1.11 | 1.81 | 2.61 | 1.66 | 1.71 | 2.43 | 1.89 |
| BigAug Zhang et al. (2020) | 1.13 | 1.78 | 4.01 | 1.25 | 1.92 | 1.89 | 2.00 |
| SAML Liu et al. (2020) | 1.08 | 1.56 | 2.49 | 1.42 | 2.01 | 1.87 | 1.73 |
| FedDG Liu et al. (2021a) | 1.32 | 1.68 | 2.32 | 1.37 | 2.18 | 1.95 | 1.80 |
| DoFE Wang et al. (2020) | **0.92** | 1.44 | 2.88 | 1.46 | 1.92 | 1.63 | 1.71 |
| RAM Zhou et al. (2022b) | 1.09 | 0.83 | 2.32 | 1.32 | **1.64** | 1.21 | 1.40 |
| SgCG | **0.92** | **0.74** | **2.17** | **1.02** | 1.67 | **0.93** | **1.24** |

important to note that, since most models provide complete code only for the Fundus dataset, we encounter discrepancies when attempting to reproduce the performance of comparative models on the Prostate dataset using the parameters outlined in the original papers. The Dice coefficient results for different models on the Prostate dataset are presented in Table 3, while the experimental results for the Average Surface Distance (ASD) are shown in Table 4. As shown in the tables, SgCG (Ours) outperforms all baseline models (SOTA) in nearly all areas. In the Dice coefficient metrics presented in Table. 3, SgCG (Ours) averages 0.68% higher than RAM (baseline). In the ASD metrics shown in Table. 4, SgCG (Ours) reduces the value by 0.16% compared to RAM (baseline).

Table 5: Ablation Study of key components in our method on Fundus Segmentation Task (%). We mark the top results in bold.

| RAM | SgCG | Dice coefficient | | | | |
|-----|------|----------|----------|----------|----------|------|
| | | Domain 1 | Domain 2 | Domain 3 | Domain 4 | Avg. |
| ✓ | - | 85.48/95.75 | 78.82/89.43 | 87.44/94.67 | 85.84/94.10 | 88.94 |
| ✓ | ✓ | **86.65/95.92** | **80.40/90.39** | **87.48/95.07** | **86.27/94.18** | **89.55** |
| | | Average Surface Distance (ASD) | | | | |
| ✓ | - | 16.05/7.12 | 14.01/13.86 | 9.02/7.11 | 8.29/7.06 | 10.32 |
| ✓ | ✓ | **14.62/7.31** | **12.96/12.21** | **8.46/6.57** | **7.54/6.64** | **9.54** |

Table 6: Ablation Study of key components in our method on Prostate Segmentation Task (%). We mark the top results in bold.

| RAM | SgCG | Dice coefficient | | | | | |
|-----|------|----------|----------|----------|----------|----------|------|
| | | Domain 1 | Domain 2 | Domain 3 | Domain 4 | Domain 5 | Domain 6 | Avg. |
| ✓ | - | 87.56 | 89.35 | 86.88 | 87.34 | **86.98** | 88.02 | 87.68 |
| ✓ | ✓ | **87.96** | **90.42** | **87.23** | **89.17** | 86.78 | **88.32** | **88.34** |
| | | Average Surface Distance (ASD) | | | | | |
| ✓ | - | 1.09 | 0.83 | 2.32 | 1.32 | **1.64** | 1.21 | 1.40 |
| ✓ | ✓ | **0.92** | **0.74** | **2.17** | **1.02** | 1.67 | **0.93** | **1.24** |

## 3.5 FURTHER PERFORMANCE ANALYSIS

### 3.5.1 ABLATION STUDY

We conduct ablation studies on the two components of SgCG to better demonstrate our contributions: the RAM module and the Semantic-guided Contrastive Generalization module. Table 5 and Table 6 present the segmentation performance of different variants of SgCG for the Funds and Prostate datasets. We observe that utilizing the contrastive generalization module consistently improves overall performance. This enhancement increases the similarity of pixel features with their corresponding semantic concepts and boosts discrimination power for mismatched pairs. Moreover, our contrastive generalization method equipped with a simple self-supervised learning strategy can further boost the performance.

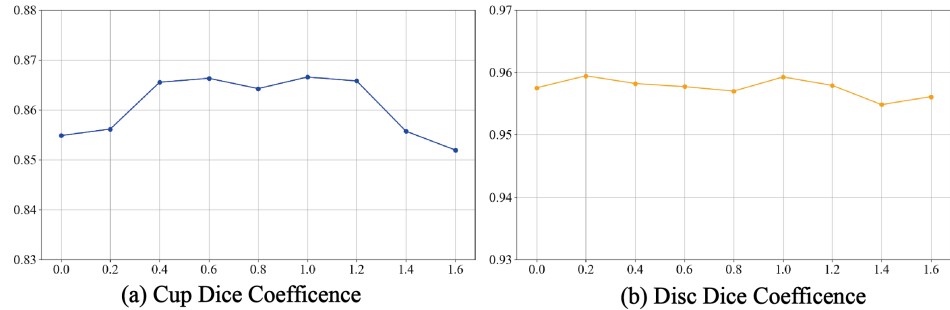

(a) Cup Dice Coeffience    (b) Disc Dice Coeffience

Figure 3: Hyperparameter analysis experiments were carried out on Funds and Prostate data sets respectively. (a) is the result of the Funds dataset, and (b) is the result of the Prostate dataset.

### 3.5.2 PARAMETER ANALYSIS EXPERIMENT

To validate the robustness of our model concerning parameter sensitivity, we conducted an analysis of the hyper-parameter $\lambda$. This analysis, performed on the Funds dataset and generalized across four domains, is illustrated in Figure 3. The results indicate that the model maintains good accuracy within a certain range of parameter values, confirming that our model's $\lambda$ demonstrates robustness and insensitivity to variations.

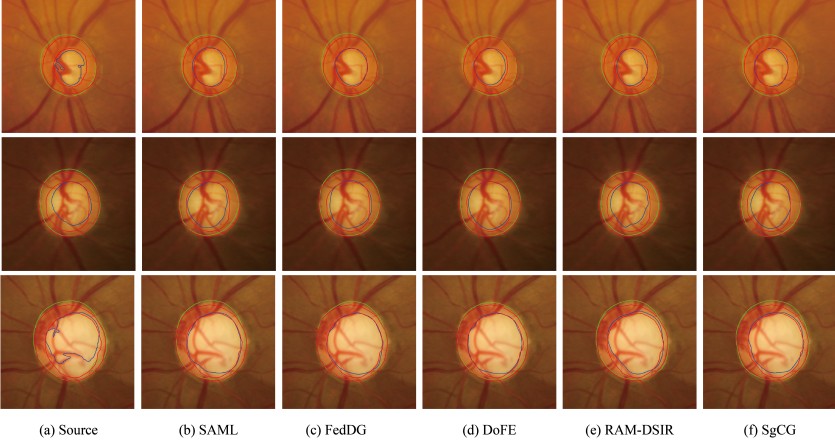

(a) Source    (b) SAML    (c) FedDG    (d) DoFE    (e) RAM-DSIR    (f) SgCG

Figure 4: Visualization of the segmentation results for the Fundus dataset. The red contours represent the boundaries of the ground truth, while the green and blue contours indicate the predictions.

### 3.5.3 VISUALIZATION OF THE SEGMENTATION

To further demonstrate the effectiveness of our method, we provide visualizations of the segmentation experiments on the Fundus and Prostate datasets in Figure 4 and Figure 5. It is clear that our method accurately segments the target structures in unseen domain images, producing smoother boundaries compared to other methods that may fail to achieve this. There is some evidence that a well-structured pixel embedding space provides the best of both worlds: reducing distribution shift, plus promoting the source task.

### 3.5.4 CONVERGENCE ANALYSIS

The convergence results of our improvement $\mathcal{L}_{feat}$ across four domains on the Funds dataset are shown in Figure 6 (a) and (b). The left side of the figure displays our loss, while the middle shows the changes in the Dice coefficient for the Optic Cup/Disc Segmentation of our model on the Funds dataset. The right side presents the changes in Average Surface Distance (ASD) for the same seg-

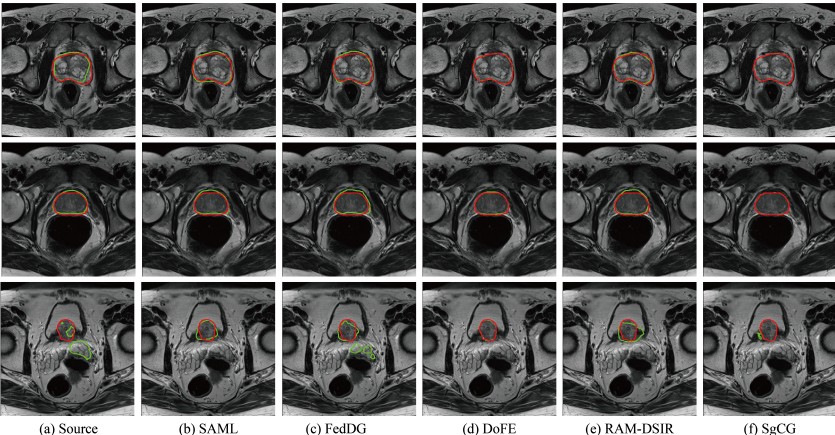

(a) Source    (b) SAML    (c) FedDG    (d) DoFE    (e) RAM-DSIR    (f) SgCG

Figure 5: Visualization of the segmentation results for the Prostate dataset. The red contours represent the boundaries of the ground truth, while the green contours indicate the predictions.

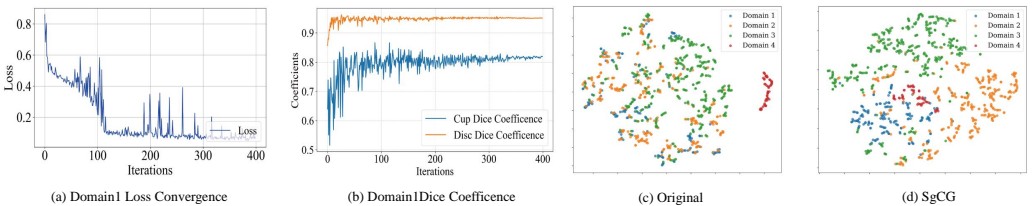

(a) Domain1 Loss Convergence    (b) Domain1Dice Coefficence    (c) Original    (d) SgCG

Figure 6: (a) Loss Convergence and Dice Coffeicence obtained by Domain1 experiment on the Funds dataset. (b) A visualization analysis was performed on the Fundus datasets experiment.

mentation task. It is evident from the figure that our loss rapidly converges with the improvement in model performance, demonstrating the effectiveness of our approach. The performance presents a show better-shaped curve and outperforms state-of-the-art significantly on the corresponding tasks.

### 3.5.5 T-SNE VISUALIZATION

To enhance our understanding and intuition, we employ t-SNE visualization Van der Maaten & Hinton (2008) to graphically represent the learned representations obtained from the SgCG method, as depicted in Figure 6 (c) and (d). This comparison is made against the original method to highlight differences. The process begins with the random selection of an image from the source domain. Subsequently, we project its high-dimensional latent features onto a two-dimensional plane. Through these t-SNE visualizations, it becomes evident that the representations derived from the SgCG method form distinct clusters. This observation underscores the method's ability to effectively discriminate between different features, showcasing the power of contrastive generalization.

## 4 CONCLUSION

In this paper, we present a novel semantic-guided contrastive generalization for medical image segmentation. To improve the generalization in DG segmentation, we introduce a particular contrastive loss at pixel label, which implicitly involves the joint learning of an infinite number of similar/dissimilar pixel pairs for each pixel-wise representation of the source domains. Finally, we get an error bound on this formulation which can assess the extent of our approach to the practical applications. Our method can successfully adapt the segmentation model to the unseen target domain through pixel-wise alignment guided by semantic distributions. The experimental results demonstrate the superiority of SgCG on various benchmarks.

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

## A APPENDIX

### A.1 RELATED WORK

In the area of medical image segmentation Elnakib et al. (2011); Shen et al. (2010), domain generalization (DG) Zhou et al. (2022a); Li et al. (2017; 2021); Qiao et al. (2020); Muandet et al. (2013) has emerged as a critical area of research, focusing on developing models that can generalize well across different datasets and imaging environments. For a new segmentation problem, models are typically from scratch, requiring substantial design and tuning. Existing DG methods mainly include meta-learning methods Liu et al. (2021b;a; 2020), feature-based methods Wang et al. (2020); Song et al. (2022) and data-based methods Zhao et al. (2019); Zhang et al. (2023); You et al. (2024).

Meta-learning divides a set of source domains into meta-train and meta-test subsets, employing meta-optimization to iteratively update model parameters, thereby enhancing performance on the meta-test subset and simulating the scenario of inferring on unseen domains. Liu et al.Liu et al. (2021b) incorporate designed constraints into the gradient-based meta-learning approach, enabling the model to extract robust anatomical features useful for predicting segmentation masks in a semi-supervised manner. FedDG Liu et al. (2021a) introduces a novel problem setting for federated domain generalization and presents an innovative approach that utilizes continuous frequency space interpolation alongside a boundary-oriented episodic learning scheme. SAML Liu et al. (2020) employs a shape-aware meta-learning strategy to enhance model generalization in prostate MRI segmentation. However, the meta-optimization process is highly time-consuming, as it requires considering all potential splitting results of the meta-train and meta-test subsets during training.

Feature-based approaches utilize domain-adaptive feature calibration or learn domain-invariant features to address domain generalization. DoFE Wang et al. (2020) introduces a novel domain code prediction branch and learning strategy to measure the similarities between input test images and various source-domain data, facilitating domain-oriented feature embedding. GLFRNet Song et al. (2022) proposes two innovative modules: a global feature reconstruction module and a local feature reconstruction module, aimed at addressing the issues of insufficient global context feature extraction and spatial information restoration within encoder-decoder networks. However, these methods do not explicitly obtain domain-invariant features for domain generalization, nor do they effectively separate features into purely domain-specific and domain-invariant representations, which limits their performance in this area.

Data-based approaches typically employ various data augmentation strategies to enhance the model's generalizability. Zhao et al.Zhao et al. (2019) introduced a learning-based method for data augmentation, demonstrating its effectiveness in one-shot medical image segmentation. Zhang et al.Zhang et al. (2023) utilized the Segment Anything model to augment image inputs for commonly used medical image segmentation models. MONA You et al. (2024) established a set of objectives that significantly enhance segmentation quality. However, the effectiveness of data augmentation largely depends on its ability to cover the data distribution in unseen domains, necessitating empirical settings and potentially data-specific modifications.

### A.2 COMPARISON METHODS AND SETTINGS

To better validate our results, we compared our outcomes with several models, including **Source**, **JiGen** Carlucci et al. (2019), **BigAug** Zhang et al. (2020), **SAML** Liu et al. (2020), **FedDG** Liu et al. (2021a), **DoFE** Wang et al. (2020), and **RAM** Zhou et al. (2022b).

**Source** Zhou et al. (2022b) model was trained using all source domain data with U-Net.

**JiGen** Carlucci et al. (2019) learns semantic labels in a supervised manner and broadens its understanding of the data through a self-supervised domain generalization approach that solves a jigsaw puzzle using self-supervised signals on the same images.

**BigAug** Zhang et al. (2020) is a data augmentation-based deep stacked transformation method for domain generalization.

**SAML** Liu et al. (2020) is a gradient-based meta-learning approach that explicitly simulates domain transfer through virtual meta-training and meta-testing during training.

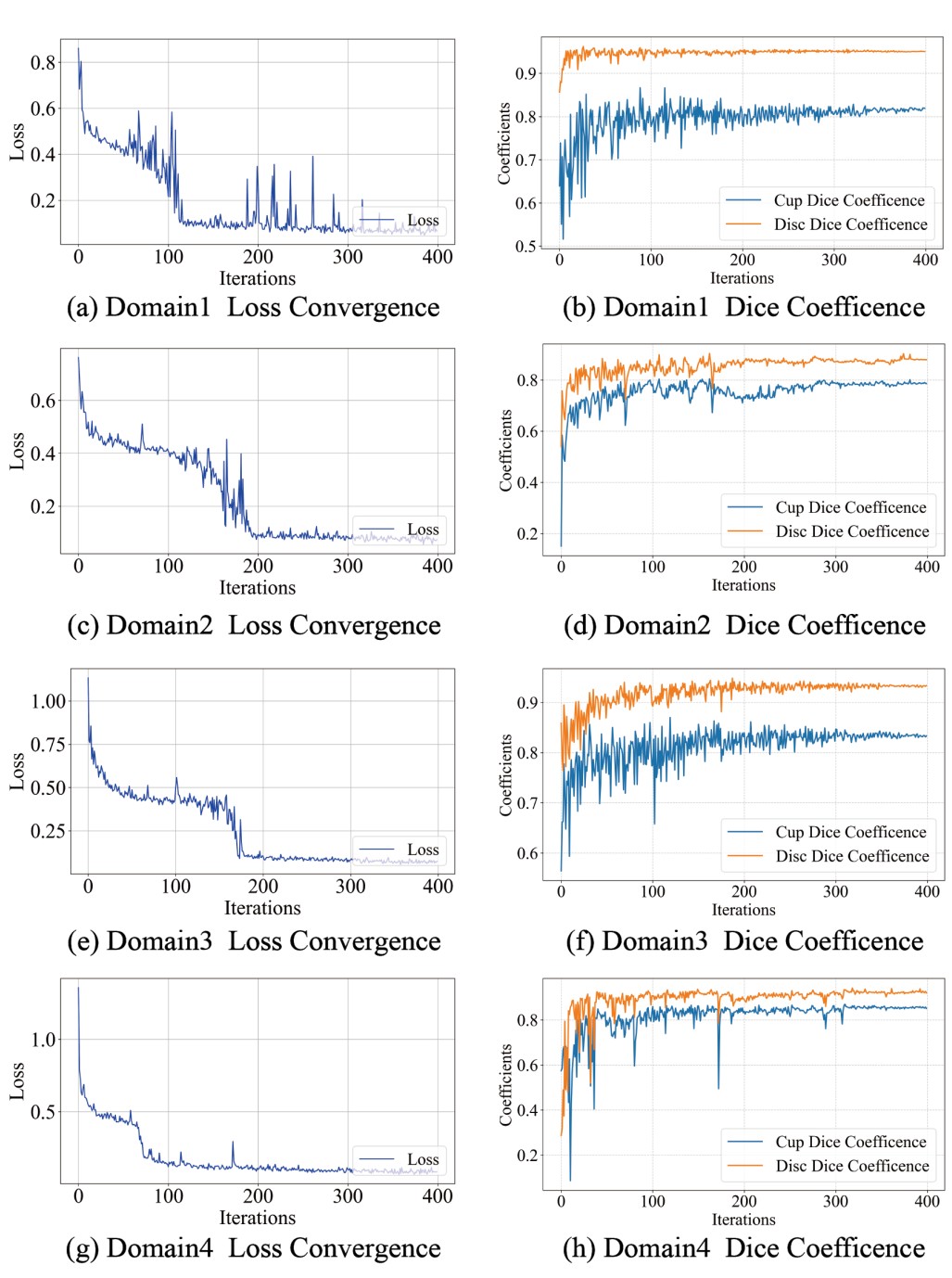

Figure 7: Loss Convergence and Dice Coefficient obtained by all domain on the Funds dataset

**FedDG** Liu et al. (2021a) is a generalizable method suitable for medical image segmentation, utilizing an effective continuous frequency space interpolation mechanism and a boundary-focused scenario learning paradigm.

**DoFE** Wang et al. (2020) introduces a domain knowledge base to learn and remember prior information extracted from multiple source domains, employing domain-focused aggregated features to enhance the domain-invariant feature representation of the original image features.

**RAM-DSIR** Zhou et al. (2022b) integrates the segmentation model with a self-supervised domain-specific image restoration (DSIR) module, designed as a multi-task paradigm, along with a Random Amplitude Mixing (RAM) module to combine low-frequency information from images across different domains for generalizable medical image segmentation domain generalization.

