# OpenReview forum: "SgCG: Semantic-guided Contrastive Generalization for  Medical Image Segmentation"
_ICLR.cc/2025/Conference — ICLR 2025 Conference Withdrawn Submission_

### Official Review · Reviewer_Bxcp · 2024-10-27

**Soundness:** 2
**Presentation:** 3
**Contribution:** 2
**Rating:** 3
**Confidence:** 3

**Summary:**

This paper introduces SgCG, a framework for domain generalization semantic segmentation of medical images. Specifically, they introduce a contrastive learning loss which inspired by semantic data augmentation to enhance source feature augmentations and generalization. Extensive experiments on two medical benchmarks showcase the effectiveness of SgCG.

**Strengths:**

The idea of the proposed SgCG seems to be simple and effective. The paper is well-written and easy to follow.

**Weaknesses:**

1. **Lack of novelty**. Contrastive learning has been widely utilized in domain adaptation [1] and generalization [2], including approaches like prototype contrastive learning that closely resemble the proposed SgCG. The authors are encouraged to  expand the related work section to include discussions on domain-adaptive and -generative contrastive learning. Emphasizing unique features of their approach that differentiate it from existing methods will strengthen the manuscript.

2. **Lack of strong baseline**. The best method RAM in comparison comes from 2022. To enhance the comparison, the authors should incorporate more recent methods in their evaluation. Meanwhile, as previously suggested, it is also essential to compare their approach with existing contrastive learning techniques to effectively showcase its contributions.

3. **Ambiguous setup**. In L88 and Eq.12, the authors imply the use of target features during training, which relates to multi-source domain adaptation. However, the experiments conducted focus on multi-source domain generalization. Clarifying this distinction and its implications for the experimental design is necessary for better comprehension.

[1] Semi-Supervised Semantic Segmentation with Pixel-Level Contrastive Learning from a Class-wise Memory Bank

[2] Calibration-Based Multi-Prototype Contrastive Learning for Domain Generalization Semantic Segmentation in Traffic Scenes

**Questions:**

Please see the weakness section.

---

### Official Review · Reviewer_fnau · 2024-10-31

**Soundness:** 3
**Presentation:** 3
**Contribution:** 2
**Rating:** 3
**Confidence:** 5

**Summary:**

This paper proposed a contrastive learning-based model for domain generalization in medical image segmentation. The key idea is to construct latent distributions for each class and use contrastive losses to improve performance. Overall, the experiment results on two public datasets have shown performance gains over the baseline model and several existing works.

**Strengths:**

Reasonable designs. Either using prototype learning or applying contrastive learning is reasonable and should improve segmentation performance for domain generalization.

Good performance. The proposed designs can achieve better performance and have been verified by corresponding experiments.

**Weaknesses:**

Technical Novelty:
This paper is closely related to ISDA. Constructing latent class distributions and conducting intra-class alignment or inter-class separation are not novel techniques in medical image segmentation. The paper seems to combine existing methods, with its task also connected to approaches in semi-supervised and self-supervised learning. See:

[1] Regularizing Deep Networks with Semantic Data Augmentation, TPAMI 2021
[2] Semi-Supervised Semantic Segmentation with Pixel-Level Contrastive Learning from a Class-wise Memory Bank, ICCV 2021
[3] Towards Generic Semi-Supervised Framework for Volumetric Medical Image Segmentation, NeurIPS 2023

Baseline Model:
The paper utilizes the RAM model but lacks a comprehensive introduction to the model and its loss function, which are critical for clarity. Additionally, as RAM was originally designed for image restoration, the choice of this model for segmentation tasks requires further justification.

Method:
While the distribution seems to be calculated by GMM in the feature space, Figure 2 does not clearly illustrate this part. Compared to ISDA, the explanation here is quite complex, particularly regarding the formulation of a general loss function by introducing infinite samples. This section would benefit from clearer and more concise exposition.

Motivation:
The paper does not introduce new insights to address the core challenge of domain generalization. For medical applications, it lacks task-specific points as well and does not offer clinical insights, which limits its relevance to medical image analysis.

**Questions:**

See the above weaknesses.
Overall, please
1. Improve the method description.
2. Show clear motivation and justify the unique contribution compared to existing works.
3. Show why it works well in the medical domain.

---

### Official Review · Reviewer_jQ5B · 2024-11-03

**Soundness:** 2
**Presentation:** 2
**Contribution:** 1
**Rating:** 1
**Confidence:** 5

**Summary:**

The paper proposes Semantic-Guided Contrastive Generalization (SgCG) to address the challenge of domain generalization in medical image segmentation. SgCG leverages contrastive learning at the pixel level, using semantic distributions to create domain-invariant features. By enhancing intra-class compactness and inter-class separability, the method aligns features from different domains to improve generalization. SgCG demonstrates state-of-the-art performance on two public datasets, Fundus and Prostate, outperforming existing methods in Dice coefficient and Average Surface Distance (ASD) metrics.

**Strengths:**

- The application of pixel-level contrastive learning guided by semantic distributions is interesting. This approach improves segmentation by encouraging domain invariance without requiring explicit target domain data.

- The paper evaluates SgCG using Dice and ASD metrics across two benchmarks.

**Weaknesses:**

The methodology is the same as that in [1*]. All the theory analysis and formulations are the same.

The figures are similar to [2*].

[3*] also tackles domain generalization in medical images, focusing on learning domain-invariant features through semantic distribution-level adaptations.

Although the manuscript shares high similarity with these three papers, none of them have been cited.

[1*] Li S, Xie B, Zang B, et al. Semantic distribution-aware contrastive adaptation for semantic segmentation[J]. arXiv preprint arXiv:2105.05013, 2021.

[2*] Wang M, Yuan J, Qian Q, et al. Semantic data augmentation based distance metric learning for domain generalization[C]//Proceedings of the 30th ACM international conference on multimedia. 2022: 3214-3223.

[3*] Guo X, Liu J, Yuan Y. Infproto-Powered Adaptive Classifier and Agnostic Feature Learning for Single Domain Generalization in Medical Images[J]. International Journal of Computer Vision, 2024: 1-24.

**Questions:**

See the weakness section.

---

### Note · Authors · 2024-11-12

I have read and agree with the venue's withdrawal policy on behalf of myself and my co-authors.